# Automated Synthesis of [^18^F]Flumazenil Application in GABA_A_ Receptor Neuroimaging Availability for Rat Model of Anxiety

**DOI:** 10.3390/ph16030417

**Published:** 2023-03-09

**Authors:** Shiou-Shiow Farn, Kai-Hung Cheng, Yuan-Ruei Huang, Shih-Ying Lee, Jenn-Tzong Chen, Kang-Wei Chang

**Affiliations:** 1Division of Isotope Application, Institute of Nuclear Energy Research, Atomic Energy Council, Taoyuan 325207, Taiwan; 2Taipei Neuroscience Institute, Taipei Medical University, Taipei 110301, Taiwan; 3Laboratory Animal Center, Taipei Medical University, Taipei 110301, Taiwan

**Keywords:** flumazenil, GABA_A_ receptor, fear conditioning

## Abstract

Clinical studies have demonstrated that the γ-aminobutyric acid type A (GABA_A_) receptor complex plays a central role in the modulation of anxiety. Conditioned fear and anxiety-like behaviors have many similarities at the neuroanatomical and pharmacological levels. The radioactive GABA/BZR receptor antagonist, fluorine-18-labeled flumazenil, [^18^F]flumazenil, behaves as a potential PET imaging agent for the evaluation of cortical damage of the brain in stroke, alcoholism, and for Alzheimer disease investigation. The main goal of our study was to investigate a fully automated nucleophilic fluorination system, with solid extraction purification, developed to replace traditional preparation methods, and to detect underlying expressions of contextual fear and characterize the distribution of GABA_A_ receptors in fear-conditioned rats by [^18^F]flumazenil. A carrier-free nucleophilic fluorination method using an automatic synthesizer with direct labeling of a nitro-flumazenil precursor was implemented. The semi-preparative high-performance liquid chromatography (HPLC) purification method (RCY = 15–20%) was applied to obtain high purity [^18^F]flumazenil. Nano-positron emission tomography (NanoPET)/computed tomography (CT) imaging and ex vivo autoradiography were used to analyze the fear conditioning of rats trained with 1–10 tone-foot-shock pairings. The anxiety rats had a significantly lower cerebral accumulation (in the amygdala, prefrontal cortex, cortex, and hippocampus) of fear conditioning. Our rat autoradiography results also supported the findings of PET imaging. Key findings were obtained by developing straightforward labeling and purification procedures that can be easily adapted to commercially available modules for the high radiochemical purity of [^18^F]flumazenil. The use of an automatic synthesizer with semi-preparative HPLC purification would be a suitable reference method for new drug studies of GABA_A_/BZR receptors in the future.

## 1. Introduction

Anxiety disorders, which are characterized by excessive worries, motor tension, and fatigue, can be clinically subdivided into several categories, including generalized anxiety disorder, panic disorder, social anxiety, agoraphobia, post-traumatic stress disorder, and obsessive-compulsive disorder [1,2,3,4,5]. Anxiety and fear produce similar behavioral responses, including increased vigilance, freezing, hypoactivity, elevated heart rate, and suppressed food consumption [6,7]. 

Generally, the goal of fear and anxiety research is to identify the neurobiological mechanisms of brain pathogenesis, and understand how to treat the potentially devastating effects of anxiety disorders in humans [8]. Numerous animal models have been used to investigate anxiety-related disorders in rodents, including paradigms based on exploration (open field, hole-board, elevated plus maze, light-dark, social interaction, mirror chamber) and fear conditioning [5,6,7,9,10,11,12]. Fear conditioning is a form of learning, in which an initially neutral conditional stimulus (CS), such as a tone, is paired with a fear-inducing, aversive unconditional stimulus (US), usually a foot shock. This results in the expression of a fear response in the presence of a conditioned stimulus alone [13,14,15,16]. Because fear conditioning occurs very rapidly and has a lasting effect, this model has been extensively investigated to explore the cellular-molecular mechanisms of learning and memory [9,10,11,12]. 

The neurobiological source of anxiety spectrum disorders is yet to be fully understood; however, a growing body of evidence suggests that the γ-aminobutyric acid (GABA) system is believed to play a key role [1,2,6,7,17]. Patients with panic disorders were found to have consistently lower brain levels of GABA than healthy controls. Some treatments of anxiety disorders will enhance GABA activity, primarily by increasing the release of non-synaptic GABA from glia, for a variety of psychiatric disorders (pain disorder, social phobia, generalized anxiety disorder, and post-traumatic stress disorder). GABA_A_ receptors are widely distributed in the brain. The majority of GABA_A_ receptors are composed of two α subunits, two β subunits, and one γ subunit in numerous combinations, which may confer different physiological and pharmacological activities [18,19]. During neurotransmission, GABA acts at inhibitory synapses through allosteric interactions with GABA_A_ receptors, and allows the opening of a Cl^−^ ion channel, thereby increasing the conductance of Cl^−^ [18]. The active site of the GABA_A_ receptor is the binding site for GABA, and can be targeted by several drugs, such as muscimol, gaboxadol, and bicuculline [20,21,22]. GABA_A_ receptors also contain several allosteric binding sites that indirectly modulate the activity of the receptor, including the clinically important benzodiazepine-binding site [23]. 

Previous preclinical studies have suggested that increased synaptic GABA concentrations enhance the affinity of GABA_A_ receptors for benzodiazepine ligands, such as flumazenil (ethyl 8-fluoro-5-methyl-6-oxo-4H-imidazo[1,5-a][1,4]benzo-diazepine-3-carboxylate, FMZ), via a conformational change (termed the “GABA shift”) [24]. Such an increase in the binding of a GABA_A_ benzodiazepine-receptor site-specific PET radio-ligand (e.g., [^3^H]-flumazenil and [^11^C]-flumazenil) has been confirmed to help with anxiety disorders at the GABA_A_ receptor level [1,2,25]. Preparation of [^11^C]-flumazenil, a confirmed benzene diazo salt-specific antagonist, began in 1981, and the compound is widely used in clinical practice since 2008 [26,27,28,29]. [^18^F]flumazenil, with an identical structure to [^11^C]flumazenil, was shown to have pharmacokinetic, peripheral metabolism, and PET imaging characteristics similar to [^11^C]flumazenil in a cynomolgus monkey [30,31]. [^18^F]flumazenil PET has evolved to be one of the most useful radiopharmaceuticals for the PET imaging of GABA_A_ receptors, with potential as a clinical tool for analyzing localization of the epileptogenic zone in pre-surgical evaluations of drug-resistant temporal lobe epilepsy (TLE), with excellent sensitivity and anatomical resolution [25,32,33].

Fear-related emotional reactions are mediated by complex neural circuitry. Recent brain imaging studies support the thought that GABA transmission in the amygdala is central to the processing of fear and anxiety [12,34,35]. Patients with damage to the amygdala exhibit deficits in fear conditioning. The amygdala also plays a key role in the defensive behavior of rodents [36]. Conditioning fear in animals is considered to be a good strategy to investigate mechanisms and pathogenesis of human anxiety disorders. To date, the neurocircuitry of conditioning has been studied primarily through lesion or inactivation experiments in rodents and, to a lesser degree, using in vivo imaging, autoradiography or histochemistry techniques [37,38,39].

This study automated the synthesis of [^18^F]flumazenil and applied the product for anxiety animal model in vivo neurologic imaging. We were able to exploit the gradient mobile phase of the high-performance liquid chromatography (HPLC) to enhance the purity of [^18^F]flumazenil, to achieve high yield radiochemical pure results, and use [^18^F]flumazenil for NanoPET/CT imaging and ex vivo autoradiography to characterize the distribution of GABA_A_ receptors in a rat model of anxiety. We expect that these results will promote straightforward labeling with, and purification of, [^18^F]flumazenil, and provide more insight into the properties of GABA_A_ receptors in vivo, and their relevance in anxiety disorders.

## 2. Results

### 2.1. [^18^F]Flumazenil Radiosynthesis and Quality Controls

All synthesis and purification procedures were performed on a Tracerlab FX-FN automated synthesizer (Figure 1A,B). [^18^F]flumazenil was produced from the nitroflumazenil precursor via a standard nucleophilic substitution fluorination reaction, as shown in Figure 1A. The produced activity of the radioligand was 0.8–2.3 GBq, with a radiochemical yield (RCY) of 15–20% decay, corrected at the end of bombardment (EOB); the total synthesis time was 90 min. This procedure resulted in a higher quality [^18^F]flumazenil, with a radiochemical purity > 95% after radio-HPLC and radio-thin layer chromatography (TLC). In an amber glass vial, the radiochemical purity of [^18^F]flumazenil in 30% ethanol remained > 90% at 8 h after the end of synthesis (EOS) at room temperature (Figure 1C,D).

### 2.2. Behavioral Study

Fear-conditioned rats received 1 to 10 tone-foot-shock trainings, each training consisting of a 20 s tone (CS) of 80 dB, and a foot-shock of 1.2 mA (US) administered for 3 s (Figure 2A). The behavioral study of freezing behavior between the closed arm of 1 and 2 using different foot-shock (0.6 and 1.2 mA) and repeat times were listed in (Appendix A). The rats who received 10 CS-US presentations froze significantly more than control animals (*t*-test: *p* < 0.001; Figure 2B). 

The effect of the number of CS-US pairings was also elucidated. The rats exhibited high levels of freezing with successive foot-shock presentations, indicating a comparable level of learning during training. Thus, we settled on a 10-trial procedure to establish an animal model of anxiety. Inspection of individual behavioral data showed considerable variability within each group. Therefore, we selected the best responders for subsequent imaging and autoradiographic analyses.

### 2.3. Ex Vivo Autoradiography

We performed ex vivo autoradiography to assess GABA_A_ receptor availability in fear-conditioned rats. In the autoradiograms of sections taken 30 min after injection of [^18^F]flumazenil, the specific localization of GABA_A_ receptors in each rat brain was examined (Figure 3). The analysis of GABA_A_ receptor distribution in rat brains indicated that the prefrontal cortex, cortex, hippocampus, and amygdala, which are known to have high densities of GABA_A_ receptors, displayed the most intense accumulation of [^18^F]flumazenil. Compared to control rats, there was a significantly lower cerebral accumulation of [^18^F]flumazenil in rats expressing contextual fear conditioning (Figure 3A).

Brain tissues from four regions, including the prefrontal cortex, cortex, hippocampus, and amygdala, were examined bilaterally. The specific binding ratio was obtained for all target regions using the pons as a reference region. For the region-dependent differences, normal rats displayed a markedly higher specific binding ratio than fear-conditioned rats in the prefrontal cortex, cortex, hippocampus, and amygdala (*p* < 0.05) (Figure 3B).

### 2.4. NanoPET/CT Imaging

The regional distribution of GABA_A_ receptors in the brain, as imaged by [^18^F]flumazenil PET, is shown in Figure 4. After injection of [^18^F]flumazenil, each time frame was observed in 10 min in both fear-conditioned rats and normal rats.

Individual dynamic images were automatically co-registered to each rat brain region template. ROIs, including the prefrontal cortex, cortex, hippocampus, amygdala, and pons, were extracted from a set of previously constructed regions. Finally, time-activity curves were generated by projecting predefined ROIs onto the dynamic images (Figure 5).

Based on comparisons with the reference region, the specific binding ratios in the control group were significantly higher than those in the fear-conditioned group. Time-activity curves showed persistence of high specific binding ratios of [^18^F]flumazenil in the four target regions, and conversely for the fear-conditioned rats (Figure 5). This phenomenon was even more profound in the last frame (50–60 min post-injection), where the comparison of fear-conditioned animals with controls showed the largest differences in specific binding ratios for all target regions. Across the target regions, the average specific binding ratios of [^18^F]flumazenil in control rats compared to the fear-conditioned group were 1.05 vs. 0.27 in the amygdala, 1.60 vs. 0.31 in the hippocampus, 2.83 vs. 0.78 in the prefrontal cortex, and 2.17 vs. 0.71 in the cortex, for control vs. fear-conditioned rats, respectively (Figure 5).

The density patterns of GABA_A_, visualized from the PET experiments, were in agreement with the results of the ex vivo autoradiography study.

### 2.5. Blocking Study

As shown previously, [^18^F]flumazenil passes through the blood–brain barrier and is localized in the regions where GABA_A_ receptors are concentrated. To further characterize the nature of the GABA_A_ receptor uptake, the effects of pre-treatment for 10 min with 200 µg diazepam (BZR agonist) on the distribution of [^18^F]flumazenil were evaluated, using ex vivo autoradiography and pre-treatment with 0.01–10 mg/kg diazepam for NanoPET/CT in vivo imaging. In ex vivo autoradiography, treatment with diazepam displayed a marked decrease of [^18^F]flumazenil accumulated in all brain regions (Figure 6). As expected for a receptor ligand in NanoPET/CT imaging, increasing the dose of diazepam resulted in reduced cerebral uptake of [^18^F]flumazenil. A rapid and almost complete washout of [^18^F]flumazenil was visible at 10 mg/kg diazepam, afforded in all brain regions. The specific uptake in the frontal cortex, cortex, and the hippocampus were significantly diminished by pre-treating rats with diazepam (0.01–10 mg/kg, i.v.). Pre-treatment with higher concentrations of diazepam resulted in fewer GABA_A_ receptors being available for binding, and [^18^F]flumazenil was eliminated from the brain at a rapid rate (Figure 7).

## 3. Discussion

Previously, [^18^F]flumazenil synthesis was conducted via isotopic exchange of ^19^F for ^18^F, using flumazenil as the precursor during synthesis [40,41,42]. The development of nitro-flumazenil as a precursor has resulted in high specific activity, which is essential in brain receptor imaging studies. However, a long synthesis time of approximately 75 min has been reported when using the same precursor. Moreover, after radiosynthesis, the needed purification of [^18^F]flumazenil using semi-preparative HPLC can result in low specific activity, as it is time-consuming process (50 min), and the procedure has been difficult to automate, resulting ^18^F radioactive decay [43,44]. Our study represents a straightforward carrier-free nucleophilic radio-fluorination method, using an automatic synthetic technique with solid extraction purification to replace semi-preparative HPLC, based on a fully automatic radiosynthesizer labeled with nitro-flumazenil precursor (Figure 1A). The total synthesis time of [^18^F]flumazenil was 90 min, and the obtained [^18^F]flumazenil exhibited high radiochemical purity (radio-HPLC (>90%) and radio-TLC (>90%), in Figure 1C,D), although the radiochemical end-of-synthesis (EOS) yield was approximately 15–20% lower than the reference (30%) [43,44]. Overall, we report a very convenient method for obtaining adequate yields and radiochemical purity of [^18^F]flumazenil.

The purification of crude [^18^F]flumazenil was previously carried out using acetonitrile and water as the mobile phase [31,45,46,47]. However, acetonitrile is flammable, volatile, toxic, and must be removed during the purification process. In contrast to acetonitrile, ethanol is less toxic, less expensive, and has lower disposal costs [29]. Therefore, we used ethanol instead of acetonitrile, and the purified product was directly diluted for injection. Using this radiosynthesis process, there are many promising aromatic ring compounds that could be substituted more conveniently [48,49,50]. However, as analyzed by radio-TLC, [^18^F]flumazenil demonstrated good stability, maintaining a purity of up to 90%, after 8 h in ethyl acetate:ethanol (80/20) at room temperature (Figure 1D).

In the present study, we used [^18^F]flumazenil, which has similar in vivo binding characteristics to [^11^C]flumazenil, to address the half-life disadvantage and to reduce errors during the long PET scanning time (90 min) [24]. Fear conditioning has become widely regarded as a valid experimental model for clinical anxiety disorders [13]. It is important to note that most neurobiological studies employ simple temporal pairings of CS and US to produce fear conditioning [51,52]. Previous studies have shown that fear conditioning can be rapidly instilled in animals, even after a single conditioning trial, and is usually maintained for long periods [14,53]. This study observed that by increasing the number of CS-US pairings, the animals exhibited higher freezing responses, indicating a comparable level of learning during the training. Additionally, the effect of shock intensity on learning was elucidated. The rats receiving 1.2 mA shocks exhibited increased freezing responses compared with those receiving 0.6 mA shocks, suggesting a significant impact of the shock intensity on associative memory formation [27]. In this study, we investigated GABA_A_ receptor availability using [^18^F]flumazenil PET in a rat model of anxiety. We found a significant reduction in GABA_A_ receptor availability in the amygdala, prefrontal cortex, cortex, and hippocampus in fear-conditioned animals. Our rat autoradiography results also supported the PET imaging findings.

Clinical studies have demonstrated that the GABA_A_ receptor complex plays a central role in the modulation of anxiety. Behavioral and physiological lines of evidence suggest that GABAergic transmission in the amygdala undergoes plastic changes which contribute to the control of fear and anxiety [45,46]. Neuroimaging studies in humans have indicated that some brain regions may be revealed as a conditioned-fear circuit, which includes the prefrontal cortex, anterior cingulate, and additional cortical and subcortical structures [28,40,45]. The currently available therapeutic agents that act on this receptor are effective anxiolytics; infusions of GABA or GABA_A_ receptor agonists into the amygdala decreased measures of fear and anxiety in several animal species, while infusions of GABA antagonists tended to have anxiogenic effects [46,47]. This suggests that the neuroinhibitory processes of GABA_A_ receptors may serve as target sites for drug action [2,39].

This is the first study to utilize [^18^F]flumazenil PET to visualize GABA_A_ receptor distribution in vivo in an animal model of anxiety. By utilizing [^18^F]flumazenil NanoPET imaging, we observed a global reduction in GABA_A_ receptor availability in rats expressing contextual fear compared to controls. Furthermore, the specific binding ratios of [^18^F]flumazenil were significantly reduced in the brain regions of the prefrontal cortex, cortex, amygdala, and hippocampus in fear-conditioned rats, which clearly depicts the effects of fear conditioning on GABA_A_ receptors. Our findings have demonstrated that the specific binding ratios of the GABA_A_ receptor are profoundly influenced by brain regions, such as the amygdala, prefrontal cortex, cortex, and hippocampus, suggesting that these brain regions may be involved in processing and storing conditioned fear memory.

To further characterize the nature of GABA_A_ receptor uptake, the effects of pre-treatment with 200 µg diazepam (BZR agonist) were studied in ex vivo autoradiography, and pre-treatment with 0.01–10 mg/kg diazepam for NanoPET/CT in vivo imaging. Pre-saturation studies have shown a high displaceable component for diazepam administration in the frontal cortex, cortex, amygdala, and hippocampus contrasted to [^18^F]flumazenil, indicating that the structural modification of [^18^F]flumazenil results will show in high affinity for GABA_A_/cBZRs (central benzodiazepine receptors) [23,30,31].

Associative learning in fear-conditioned animals is thought to involve different forms of activity-dependent synaptic plasticity [49]. Although previous data and the results from the present NanoPET study implicate these brain regions in contextual conditioning, the exact contribution is still unclear. This reduction in GABAergic neurotransmission may be caused by any or all of the following: loss of GABAergic inter-neurons, loss of GABA_A_ receptors, or changes to GABA_A_ receptor subunits, leading to alterations in receptor properties [27]. The most interesting result of this study is the identification and elucidation of the involvement of GABA_A_ receptors in animals in the expression of contextual fear responses. The results of this study emphasize the importance of careful study design and control of experimental conditions when imaging GABA_A_ receptors. Further studies incorporating cued fear and fear extinction as behavioral measures may elucidate the exact role of GABA_A_ receptors. We suggest that these results will facilitate the development of new treatment options for patients with anxiety disorders in the long run. 

## 4. Materials and Methods

### 4.1. [^18^F]Flumazenil Radiosynthesis

All of the chemicals were purchased from commercial vendors and were of pharmaceutical grade. [^18^F]fluoride was produced via the ^18^O (p,n) ^18^F reaction on an enriched [^18^O]water target, and irradiated with a proton beam (17 μA) using an EBCO TR30/15 cyclotron (Ebco industries, Mumbai, India). After irradiation, the nucleophilic substitution reaction based on [^18^F]F^−^/H_2_[^18^O]O was transferred to a semi-automated TRACERlab FX-FN synthesis module (GE Healthcare, Chicago, IL, USA). Aqueous [^18^F]fluoride/H_2_[^18^O]O was passed through an anion exchange cartridge (1-X8 resin, Bio-Rad) (Hercules, CA, USA) preconditioned with K_2_CO_3_. The fixed [^18^F]F− was eluted from the cartridge with a solution of Kryptofix 2.2.2 and K_2_CO_3_. After azeotropic evaporation, 10 mg of nitroflumazenil in 1 mL of dimethylformamide (DMF) was added to the [^18^F]F^−^/Kryptofix 2.2.2/K^+^ complex. Fluorination proceeded for 15 min at 150 °C. The reaction mixture was then cooled to 50 °C, diluted with 2 mL of water, and passed through a Waters Alumina-N cartridge for solid phase extraction. Then, [^18^F]flumazenil was purified via solid phase extraction and passed through the Waters Alumina-N cartridge. [^18^F]flumazenil was purified using semi-preparative HPLC (Waters μ-Bondapak C18 3.9 × 300 mm, 10 µm; mobile phase from 0 to 20 min, 20% ethanol; 20–40 min, 30% ethanol; flow rate, 3 mL/min; UV detection at 400 nm) (Milford, MA, USA). The product was collected and its radiochemical purity and stability were measured using radio-HPLC (Waters and Aligent 1100 and the Perkin Elmer’s Radiomatic 150 TR) and radio-TLC (BIOSCAN Autochanger 3000 type) (Fifth Circle, Amman, Jordan).

Quality control of the final injectable [^18^F]flumazenil solution was performed using HPLC and radio-TLC. An Agilent series 1100 HPLC (Cosmosil 5C18-MS-II C18, 5 μm, 4.6 × 150 mm) equipped with ultraviolet and radioactivity detectors (Bioscan Flow count and the Perkin Elmer’s Radiomatic 150 TR) was used to analyze the radioactive [^18^F]flumazenil, with the mobile phase (H_3_PO_4_/acetonitrile = 70:30) delivered at 1 mL/min. The identity of [^18^F]flumazenil was confirmed by comparing the retention times with those of unlabeled flumazenil. TLC was carried out on a Merck Si-60 plate, and an ethyl acetate:ethanol (80/20) solution was applied to develop the TLC plates (the Rf values were 0.05 for [^18^F]fluoride and 0.70 for [^18^F]flumazenil). The stability was investigated, and the radiochemical purity of [^18^F]flumazenil was monitored while it was stored in 30% ethanol at room temperature for 2, 4, 6, and 8 h.

### 4.2. Animals

Animal housing and experiments were approved by the Ethical Animal Use Committee of the Institute of Nuclear Energy Research Atomic Energy Council (INER), and performed in compliance with Taiwan’s laws for the care and use of laboratory animals (LAC-2019-0253). All experiments were carried out in male Sprague Dawley rats (3–4 weeks), purchased from a licensed breeder (BioLASCO Taiwan Co., Taipei, Taiwan). The rats were maintained at a room temperature of 21 ± 2 °C with 50 ± 20% humidity under an automatic 12 h light and 12 h dark cycle, with food and water provided ad libitum. All experiments were performed during the activity phase of the animals, between 9:30 and 17:30. The rats were allocated to two groups: one group used for fear conditioning studies, and the other the control group.

The experimental setup of fear-conditioned animals has been explained in detail in previous studies [54,55,56]. The rats were placed in a fear conditioning chamber with a metal grid for shock application. After a 2 min adaptation period, a tone cue was presented at 80 dB for 20 s. An electric foot-shock (1.2 mA) was administered via the metal grid during the last 3 s of the tone presentation, and co-terminated with the tone. After the shock presentation, a second identical trial preceded an inter-trial interval of 1 min. Rats were trained with 1–10 CS-US pairings to investigate the effect of the number of CS-US pairings.

The freezing behavior of the animals was recorded using a video camera. Freezing behavior was defined as immobility, except for respiration movements. Freezing responses in different conditions were measured over the entire 3 min interval and expressed as a percentage of the observation interval. Freeze (%) = (Freeze time/Total time) × 100%. Control animals underwent the same procedure but did not receive a CS-US presentation. The rats received 10 tone-foot-shock pairings and subsequently NanoPET/CT imaging and ex vivo autoradiography of their brains was performed.

### 4.3. Ex Vivo Autoradiography

The fear-conditioned and control rats (three rats per group) received an injection of 37 MBq/200 μL [^18^F]flumazenil, via the lateral tail vein, 2 h after the fear-conditioning experiments. The animals were sacrificed by the administration of carbon dioxide 30 min after the injection. The brains were rapidly removed, placed in optimal cutting temperature compound (OCT) embedding medium, and frozen with powdered dry ice. The brains were cut into 20 μm thick sagittal sections on a cryostat (LEICA CM3050S, Wetzlar, Germany), thaw-mounted on gelatin-coated microscope slides, and air dried at 25 °C. The slides containing the brain sections were exposed to a film (BAS-SR2040, Fuji Photo Film) in an autoradiographic cassette for 72 h. The optical densities of the autoradiograms were determined with an image analysis system using the FLA-5000 and MultiGauge ver 2.0 software (Fuji Photo Film, Akasaka, Minato, Tokyo, Japan).

The specific binding ratio in all target regions (prefrontal cortex, cortex, hippocampus, and amygdala) could be obtained using the pons as a reference region. Specific binding ratios were calculated using the following equation: (target region–reference region)/reference region.

### 4.4. NanoPET/CT Image Acquisition and Analysis

In an independent experiment, rats were transported to an imaging facility 2 h after receiving 10 tone-foot-shock pairings. The fear-conditioned and control rats (five rats per group) were anesthetized with 3% isoflurane gas and injected intravenously with [^18^F]flumazenil (37 MBq/200 μL) via the lateral tail vein. The apparent behavioral status and vital signs of the rats were monitored during the scan to adjust for anesthesia. The temperature of the animals was maintained at 37 °C. For the PET study, the rats were positioned in a NanoPET/CT PLUS system (Bioscan Europe, Ltd., Paris, France), using the dynamic mode for data acquisition from 0 to 60 min (a series of six frames, 6 × 10 min).

An acquired PET dynamic series was reconstructed with PET images consisting of six time frames (six frames × 10 min). The frames were then reconstructed using 3D Monte Carlo reconstruction with a matrix size of 128 × 128 pixels. Reconstructed PET data were further processed using PMOD ver 3.3 software (PMOD Technologies Ltd., Fällanden, Switzerland) and co-registered to a CT template. The PMOD v3.3 rat brain regions template was used to define the regions of interest (ROI). On the CT template, the amygdala, hippocampus, cortex, prefrontal cortex, and pons were delineated as ROIs. The PET data were matched to the respective CT data sets. The specific binding ratio in each ROI was computed for each frame, resulting in the generation of time-activity curves.

### 4.5. Blocking Study

In vivo competitive binding of 200 µg diazepam (benzodiazepine receptor agonist, BZR agonist) in the brain was investigated in normal rats using [^18^F]flumazenil labeling. After 10 min of pretreatment with diazepam (BZR agonist) through intraperitoneal injection (0.01, 0.1, 1, and 10 mg/kg), the rats received an injection of 37 MBq/200 μL [^18^F]flumazenil. The pre-treated rats were imaged using NanoPET/CT following the same procedures described above for the control rats. Occupancy (%) = (ROI of control group brain region − ROI of pretreated group brain region/ROI of control group brain region) × 100%.

### 4.6. Statistical Analysis

Data points represent mean values and error bars represent standard deviations. Statistical analyses were performed using the unpaired two-tailed Student’s *t*-test, unless otherwise noted; *p* values less than 0.05 were considered statistically significant, and significance levels in graphs are marked as follows: * *p* < 0.05, ** *p* < 0.01, and *** *p* < 0.001.

## 5. Conclusions

The straightforward radio-labeling and purification procedure described in this study can be easily adapted by commercially available modules for the high radiochemical purity of [^18^F]flumazenil. The results demonstrate that this fully automated nucleophilic [^18^F]fluorination system can replace traditional methods, and the product of [^18^F]flumazenil can be utilized in PET imaging. To quantify GABA_A_ availability in in vivo NanoPET/CT and ex vivo autoradiography in an animal model of anxiety.

## Figures and Tables

**Figure 1 pharmaceuticals-16-00417-f001:**
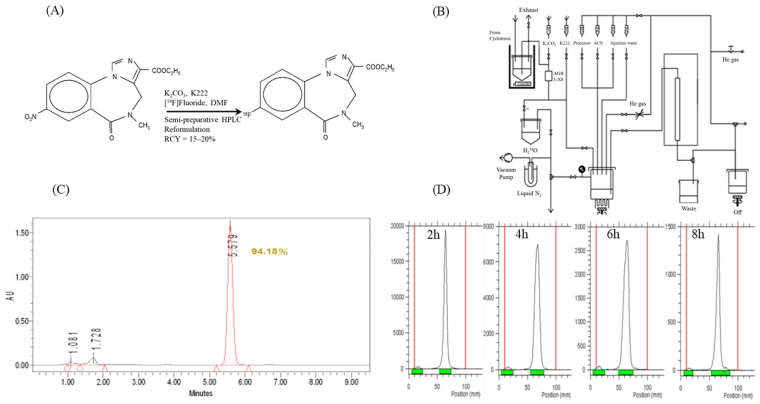
(**A**) Scheme of synthesis of [^18^F]flumazenil. The non-carrier added [^18^F]flumazenil radiofluorination reaction with tetra-butyl ammonia hydrogen carbonate phase catalyzed in DMSO nitro-flumazenil solution. (**B**) Flowchart (LabView Operational Window) for the synthesis of [^18^F]flumazenil. (**C**) The diagram of radiochemical purity of [^18^F]flumazenil. Radio-HPLC analysis of [^18^F]flumazenil using a COSMOSIL 5C18 MS-II column (4.6 × 150 mm), a flow rate of 1 mL/min, and mobile phase of H_3_PO_4_/acetonitrile 70:30. (**D**) The stability study of [^18^F]flumazenil was performed on radio-TLC, and the analysis used a Merck Si-60 plate with an ethyl acetate:ethanol (80/20) solution. Separate display at 2, 4, 6, and 8 h later. Radio-TLC was performed on a Merck Si-60 plate with an ethyl acetate:ethanol (80/20) solution.

**Figure 2 pharmaceuticals-16-00417-f002:**
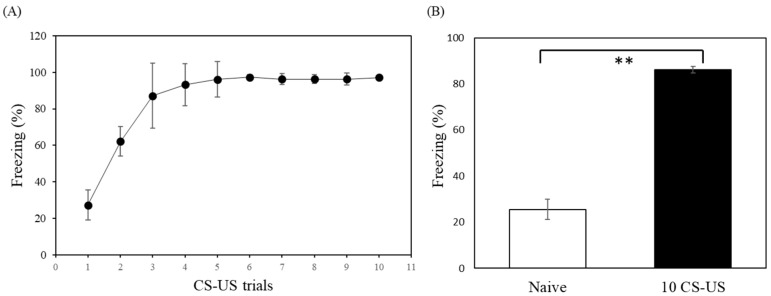
Analysis of freezing behavior in fear-conditioned and control rats. (**A**) The effect of the number of CS-US pairings. Fear-conditioned rats received 1–10 tone-foot-shock training, consisting of a 20 s tone (CS) of 80 dB, and a foot-shock of 1.2 mA (US) administered for 3 s. This study observed that increasing the number of CS-US pairings caused the animals to exhibit a higher freezing response. (**B**) The rats which received 10 CS-US presentations froze significantly more than control animals. The control group (Naive) showed without freezing than CS-US group presentated. Data points represent the mean values, and the error bars represent standard deviations. **, *p* < 0.001 (*t*-test).

**Figure 3 pharmaceuticals-16-00417-f003:**
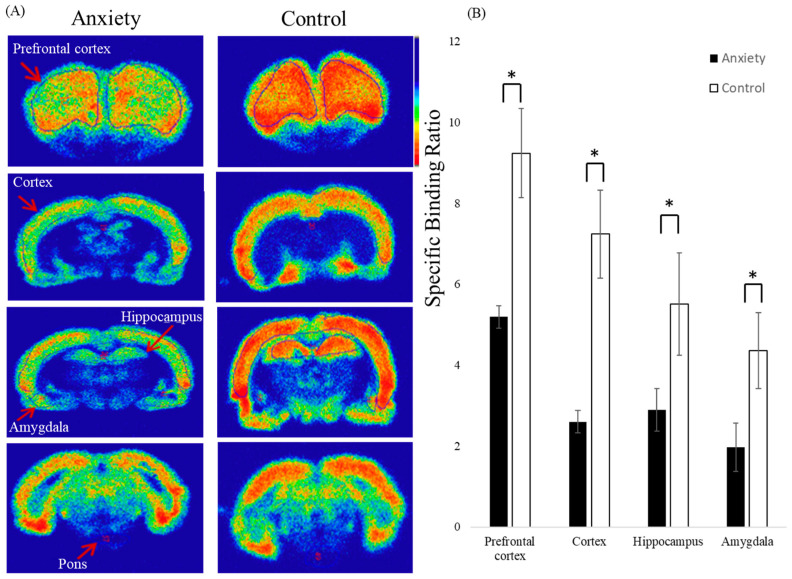
Ex vivo autoradiography of [^18^F]flumazenil in fear-conditioned (anxiety group) and control rats. The autoradiograms of sections were taken 30 min post-injection of [^18^F]flumazenil. (**A**) In comparison with control rats, the fear-conditioned rats displayed lower accumulation of [^18^F]flumazenil in the prefrontal cortex, cortex, amygdala, and hippocampus. (**B**) The normal rats displayed markedly higher specific binding ratios than the fear-conditioned rats in prefrontal cortex, cortex, hippocampus, and amygdala (*p* < 0.05, unpaired t test). Black = anxiety group, white = control group. (* *p* value < 0.05 compared to control group).

**Figure 4 pharmaceuticals-16-00417-f004:**
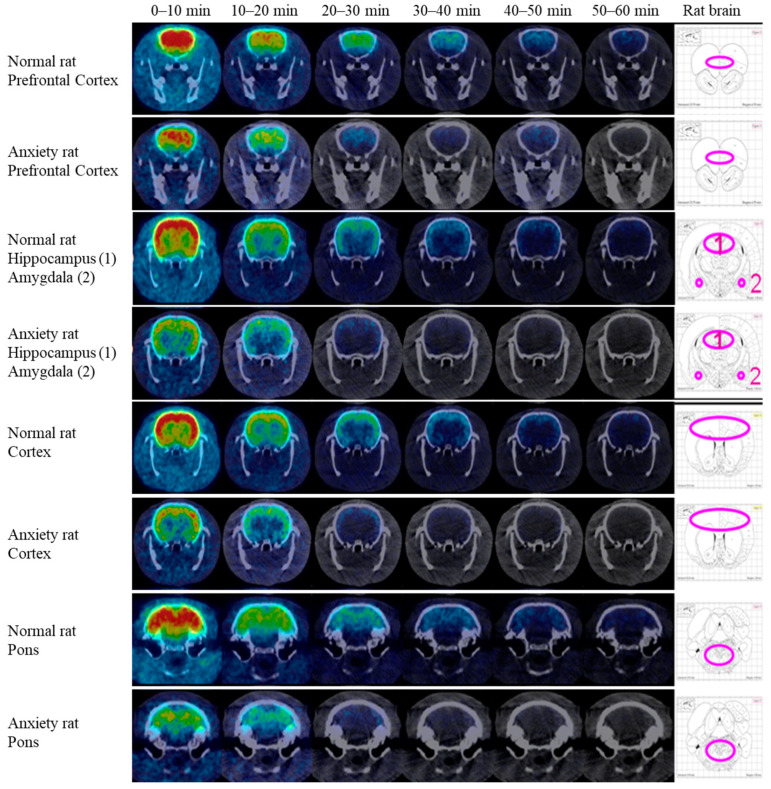
The brain regional distribution of GABA_A_ receptors as imaged by [^18^F]flumazenil NanoPET/CT in fear-conditioned (anxiety) and normal rats. The acquired NanoPET/CT dynamic series were reconstructed with NanoPET/CT images, consisting of six time frames (0–10, 10–20, 20–30, 30–40, 40–50, 50–60 min). After injection of [^18^F]flumazenil, peak uptake was observed at the first time frame (0–10 min post-injection) in both fear-conditioned rats and normal rats. In all time frames, the normal rats displayed markedly higher uptake of [^18^F]flumazenil than the fear-conditioned rats in the prefrontal cortex, cortex, amygdala, and hippocampus. The pons was delineated as ROIs and used as a reference region.

**Figure 5 pharmaceuticals-16-00417-f005:**
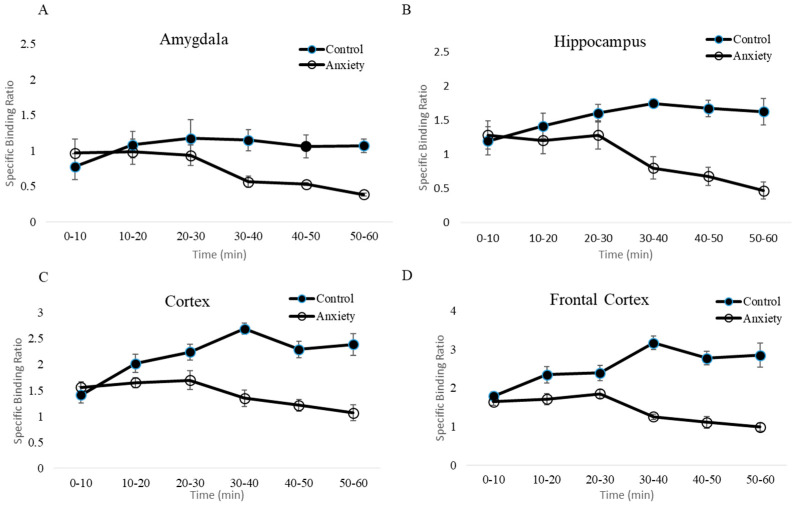
[^18^F]flumazenil time-activity curve for the fear-conditioned (anxiety) and control rats. (**A**–**D**) Separate displays of the time-activity curves from the amygdala, hippocampus, cortex and frontal cortex, respectively. The time-activity curves were generated by projecting the specific uptake ratios, calculated from the pre-defined ROIs, onto the dynamic images. Based on comparisons to the reference region, the specific binding ratios in the control group were significantly higher than those in the fear-conditioned group in the four target regions.

**Figure 6 pharmaceuticals-16-00417-f006:**
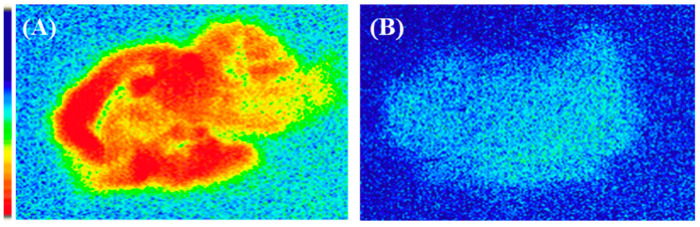
Ex vivo brain autoradiograms of [^18^F]flumazenil only (**A**) and [^18^F]flumazenil combined with 200 µg diazepam (BZR agonist) (**B**), after post-injecting, via the tail vein, [^18^F]flumazenil 37 MBq/0.2 mL for 30 min. (A) represents the binding of [^18^F]flumazenil to the high-density cortical regions of the brain, while (B) represents that the binding of [^18^F]flumazenil was blocked by the addition of diazepam.

**Figure 7 pharmaceuticals-16-00417-f007:**
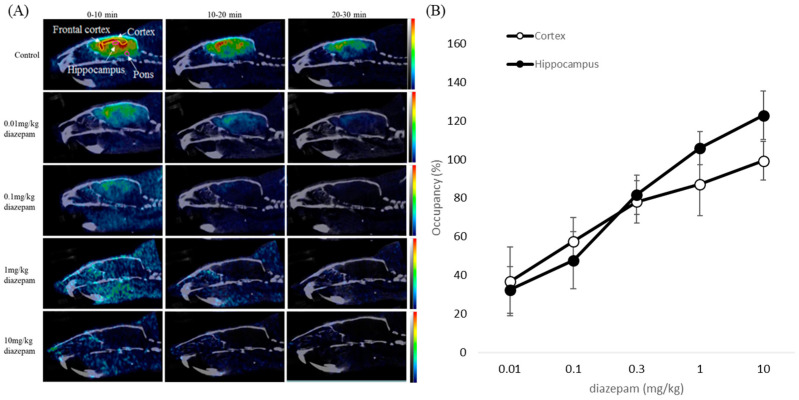
The PET images for dose-dependent inhibition of [^18^F]flumazenil by pre-treatment with diazepam (0.01, 0.1, 1, 10 mg/kg). (**A**) The specific uptake in frontal cortex, cortex, and hippocampus as showed by NanoPET/CT image. (**B**) When pre-treating rats with diazepam, the images were significantly diminished. Increasing the dose of diazepam resulted in reduced cerebral uptake of [^18^F]flumazenil. Maximal displacement was achieved within a period of 10 min post-injection of [^18^F]flumazenil.

## Data Availability

The data that support the findings of this study are available on request from the corresponding author. The data are not publicly available due to privacy or ethical restrictions.

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
