# Peer review of "Automated Synthesis of [^18^F]Flumazenil Application in GABA_A_ Receptor Neuroimaging Availability for Rat Model of Anxiety"

_pharmaceuticals, 2023, doi:10.3390/ph16030417_

Round 1

Reviewer 1 Report

The article titled Evaluation of automated synthesis of [18F]flumazenil and its application for neuroimaging of GABAA receptor availability in a  rat model of anxiety is accepted after consideration of the following.

!) Abstract, authors should mention background to their study.

2) introduction should mention recent studies.

3) the rational of this studies should improve and support  with recent publications

4) references should be updated.

5) conclusion should be improved.

6) in vivo and in vitro should be italic.

Author Response

Thanks all reviewer for the advice with scientific view.

  1. I have modified all of the abstract and whole artice for the background of this study.
  2. Add some recent reference in the introduction.
  3. In discussion also modify and add some recent reference for discussion.
  4. Reference had updated.
  5. The conclusion had updated.
  6. in vivo, ex vivo and in vitro had changed to italic.

Reviewer 2 Report

The author's experimental design is highly comprehensive, and the logic is very clear. First, there are a few minor issues that need to be addressed.

1, Figure 1 is too blurry to see clearly.

2, The iTLC data in Fig 1 looks strange.  Usually, the peak should stay at the original for successful labeling.  The author’s data seems the opposite way. It is better to run a free 18F as a control experiment. 

3, Please add a scale bar to figure 3 (A)

4, I am very interested in the BioD of the 18F flumazenil and the serum stability of 18F flumazenil. If possible, the pharmacokinetics and the metabolism of 18F flumazenil are also valuable. It is easier to get by drawing the blood of the rats and running on radio-HPLC at different time points.

Author Response

Thanks all reviewer for the advice with scientific view.

  1. We update the clear figure 1 and other figures.
  2. We have corrected the mobile buffer, and in addition file we have shown a unpurification compound of 18F. In the materials and methods of this article, we had write down the Rf values were 0.05 for F-18, and 0.7 for the F-18-FMZ.
  3. Figure 3(A) and figure 4 had add a scale bar.
  4. In the bio-distribution, pharmacokinetic, metabolism of F-18-FMZ, we had finish those study. We also study the metabolite of Flumazenil by hepatic matrix and rat liver homogenate. It is very sorry for those data will not show on this article, there are publish later.

Reviewer 3 Report

Dear Author,

Thanks for submitting your research manuscript entitled "Evaluation of automated synthesis of [18F]flumazenil and its application for neuroimaging of GABAA receptor availability in a rat model of anxiety".

Before giving my final comments as well as the final revision of this manuscript, the author needs to address the following comments scientifically.
Major concerns:

Please find out the following comments:

·         The rationale and purpose behind selecting the automated synthesis of [18F]flumazenil and its application for neuroimaging of GABAA receptor availability in anxiety is explained very poorly, irrelevant and incomplete manner throughout the manuscript.

·         Title and abstract is misleading the reader. Title needs to reframe in simply manner.

·         Lack of update as well old & outdated references with incomplete experimental design is another major concern.  

·         In Abstract direct statement “The main goal of our study was to investigate a fully automated nucleophilic fluorination system, with solid extraction purification, developed to replace traditional preparation methods and to detect un-derlying expressions of contextual fear and characterize the distribution of GABAA receptors in fear-conditioned rats is confusing. Need to reframe accordingly these types of errors throughout the manuscript.

·         This is preliminary investigation. Reason and explanation behind the updation and initiation of this now as research paper?? With these, lack of limitations in current research, especially, on the basis of current evaluation, it is tough to quote “Our rat autoradiography results also supported the findings of PET imaging. Key findings were obtained by developing straightforward labeling and purification procedures that can be easily adapted to commercially available modules for the high radio-chemical purity of [18F]flumazenil. Paper can’t be accepted and encourage to team for further proceeding the methodologies and results.

·         Rationale, Selection and evaluation of NanoPET/CT images for brain regional distribution of GABAA receptors is very poorly explained, and justified in abstract, intro as well in discussion part.

·         The reviewer found irrational and non-scientific justification in the abstract—introduction and discussion part.

·         Abstract is very poorly written and very confusing. Irrational and fused with repetitions. The reviewer found irrational and non-scientific justification in the abstract—introduction and discussion part.

-          Is this abstract or explanation of results??? What authors want to say? The incomplete justification and scientific correlation is another concern, and not matched with observed results.

-          Just focus, what you did and what you perform. Don’t add crisp in the science. The reviewer feels the author needs to elaborate and justify it with proper citations and strong evidence. The author fails to explain the relevant justification in the introduction as mentioned in the discussion part.

·         A major drawback is a lack of supporting pre-clinical and clinical evidence regarding targeting drugs.

·         Throughout the manuscript, the main focus is not clear. Complete mismatch of abstract, introduction, results and discussion.

Title:

·         Mismatch of title with relevant introduction and conclusive remarks in the conclusion part.

Abstract:

-     The rationale behind this research is not well explained, and several major concerns still constrain the reviewer's enthusiasm for publishing this manuscript.
Introduction:

- The basic literature is not well written and does not even include any literature on alternative approaches with updated references regarding involvement of current drug treatment/techniques used in pathogenesis and development of anxiety associated neurological illnesses.

- Authors fail to justify the correlation, and almost irrational and common information is present in the introduction part.

Material and methods:

-     Major drawback is the lack of supporting references and incomplete experimental and paradigms.

- All parameters are poorly explained.

- In order to support the assessment of all mentioned parameters in his study, the author should provide all the source documents and data he/she has followed for all assays and estimates.

- How was the dosing determined? Dose-responses should be performed.

- How was the sample size determined? Ideally, a priori sample size calculation should be performed to determine the appropriate sample size.
- Normality and variance homogeneity should be assessed across all groups of the same outcome variable and not individual experimental groups. If the data were not normally distributed or variance homogeneity was not met, nonparametric tests need to be performed.
Parametric data should be reported as mean +/- SD, while nonparametric data should be given/displayed as median and interquartile range. Longitudinal data should be analyzed using repeated measures tests.

Results:

-          All results are very-very poorly explained. Revised all.

-          All PET images are highly blurred, without scaling and there is no clarity for easy understanding. Not acceptable in current form. Need to provide high resolution images?

-          Re-check stat of figure 2B, 3 and 5 and confirm either statistical symbols are properly mentioned in graphs or not? I am sure, symbols are wrongly or manipulated???

-          Stat is another major concern. Need to verify. Therefore, provide the supplementary data of all graphs for further verification. Without this, article can’t proceed further.

-          Results need more clarification and significant justification. Differentiating between the outcome and the discussion sections is quite difficult.

-          High note: Must provide all results description and Use proper statistical reporting: i.e. for the results of each statistical test, the authors should report the statistical test that was applied, the test statistic (e.g. t, U, F, r), degrees of freedom as subscripts to the test statistic, and the exact probability value, including those for normality and variance homogeneity tests. Statistics should be reported in APA format, i.e.: t(df) = value, p = value; F(df1,df2) = value, p = value; r(df) = value, p = value; [chi]2 (df, N = value) = value, p = value; Z = value, p = value.  Include statements on the tests for normality and variance heterogeneity and respective results. If the data were not normally distributed or variance heterogeneity was not met, nonparametric tests need to be applied.

Discussion:

-     To address the outcome of in-vivo measures/results separately avoiding the anxiety associated neurocomplications and maintaining physiological condition and how they correlate with the existing literature, it would be better if the author restructured to take a more critical approach for effective in neurotoxic conditions.

- The explanation of all is very poor, and need to specify the scale bar properly.
-     In the discussion and the conclusion, the aims, rationale, and future perspectives are not evident clearly in relation with in-vitro and in-vivo experimentation.
-     The discussion is usually unorganized at the beginning to address all the observations and evaluate them at the end. It makes the results easier to contextualize and simpler to comprehend.

- Furthermore, a minimal critical analysis should be provided, along with current study limitations as well the future perspective as separate paragraph.

Conclusion:

-          Need to revise the conclusion in a scientific manner. Not accepted in its current form.

-          This reviewer considers that this paper cannot be published in the present form. A detailed revision shortening, ordering and following the commented ideas could improve this interesting paper in a significant manner.

-          Several typewriting mistakes are present and needing correction. This reviewer remains at entire disposal for the next version.

Author Response

Thanks reviewer's advice with seientific view.

  1. I had modified the whole article with 'track changes' function wish could improve the past shortcomings。
  2. In English editing services, in the original article had modified by an experienced scholar。 But in this version, if reviewer think it need to be reviewed again, I could immediately do the English revision assignment。
  3. If reviewer had any suggestions, please put them forward, I will reply and fixed immediately。

Round 2

Reviewer 3 Report

Dear Author, 

Reviewer surprise to see nothing in revised manuscript. You are requested to revised the manuscript as per given comments. 

Author Response

 I am sure had update my article at last time。 But I still sorry for all of reviewers do not see any addition revised manuscript in this system。 Now I resubmitted again, which all of editors could review again。  

Round 3

Reviewer 3 Report

Dear Author, 

after careful revision, revised manuscript can be proceed further for publication.